# Iron Chelation Property, Antioxidant Activity, and Hepatoprotective Effect of 6-Gingerol-Rich Ginger (*Zingiber officinale*) Extract in Iron-Loaded Huh7 Cells

**DOI:** 10.3390/plants12162936

**Published:** 2023-08-14

**Authors:** Hataichanok Chuljerm, Narisara Paradee, Dabudsawin Katekaew, Panaphat Nantachai, Kornvipa Settakorn, Somdet Srichairatanakool, Pimpisid Koonyosying

**Affiliations:** 1School of Health Sciences Research, Research Institute for Health Sciences, Chiang Mai University, Chiang Mai 50200, Thailand; hataichanok.ch@cmu.ac.th; 2Environmental-Occupational Health Sciences and Non Communicable Diseases Research Center, Research Institute for Health Sciences Chiang Mai University, Chiang Mai 50200, Thailand; 3Department of Biochemistry, Faculty of Medicine Chiang Mai University, Chiang Mai 50200, Thailand; narisara.p@cmu.ac.th (N.P.); kornvipa_s@cmu.ac.th (K.S.); somdet.s@cmu.ac.th (S.S.); 4Science Classroom Affiliated School Project, Chiang Mai University Demonstration School, Chiang Mai University, Chiang Mai 50200, Thailand; dabudsawinkatekaew@gmail.com (D.K.); panaphatmark2545@gmail.com (P.N.)

**Keywords:** ginger extract, 6-gingerol, hepatoprotective, iron

## Abstract

Iron is essential for numerous biological processes; however, an iron imbalance can contribute to a number of diseases. An excess of iron can accumulate in the body and subsequently induce the production of reactive oxygen species (ROS), leading to oxidative tissue damage and organ dysfunction. The liver, a major iron storage site, is vulnerable to this iron-induced oxidative damage; however, this issue can be overcome by the chelation of excess iron. This study aimed to investigate the effect of 6-gingerol-rich ginger (*Zingiber officinale*) extract on iron chelation, antioxidation, and hepatoprotective function in protecting against iron-induced oxidative liver cell injury. In experiments, 6-gingerol was confirmed to be a main bioactive component of the ginger extract and possessed free radical scavenging activity, decreasing ABTS^•+^ and DPPH^•^ radical levels, and inhibiting AAPH-induced red blood cell hemolysis. Interestingly, the extract significantly reduced the levels of labile cellular iron (LCI), intracellular ROS, and lipid peroxidation products (TBARS) in iron-loaded human hepatoma (Huh7) cells. In conclusion, this work highlights the iron chelation property of 6-gingerol-rich ginger extract and its antioxidant activity, which could potentially protect the liver from iron-induced oxidative tissue damage.

## 1. Introduction

For almost all living organisms, iron is a biologically essential metal that is involved in a variety of metabolic activities, such as energy production, DNA synthesis, and cellular respiration [1]. However, an imbalance in iron levels in the body has noticeable effects on health. It is widely assumed that excess iron promotes the production of reactive oxygen species (ROS) via the Fenton reaction [2,3,4]. Under iron overload conditions, the body has no physical mechanism to eliminate the excess iron; thus, it may accumulate in many vital organs, such as the liver, heart, and endocrine glands, leading to tissue damage and organ dysfunction [5]. Previous studies reported that liver damage is a major cause of death in patients with iron overload [6,7]. In addition, iron-induced ROS production promotes the expression of proinflammatory genes, which play an important role in the severity of liver diseases [8]. Antioxidant supplementation is a promising strategy to ameliorate the undesirable effects of iron overload and prolong life. Plants have been considered a major source of antioxidants because they contain a wide variety of bioactive compounds [9]. Previous studies have demonstrated that phytochemical compounds such as polyphenols, flavonoids, carotenoids, and some vitamins are powerful antioxidants that can be used to treat oxidative-stress-related diseases [10,11,12].

Ginger (*Zingiber officinale*) is widely used as a spice and herbal remedy. It contains various bioactive components, including phenols, flavonoids, terpenes, and polysaccharides. The phenolic compound is mainly attributed to gingerols, shogaols, and paradols [13,14]. Gingerol, a major (and pungent) component in ginger, exhibits various biological activities such as antioxidant, anti-inflammatory, anticancer, and antimicrobial activity [15,16]. Ginger is considered a safe medicine, as there are no significant side effects related to its usage. The accumulated studies have reported the antioxidant activity and inhibitory effects of ginger on various inflammatory cytokines. However, few studies have explored the iron chelation property of 6-gingerol in iron-induced liver injury. Therefore, this study aims to evaluate the effect of ginger extract on the iron chelation properties and antioxidant activity in the liver cells under iron overload conditions.

## 2. Results

### 2.1. Chemical Compositions of Ginger Extract

Ginger contains various bioactive constituents, including phenolic compounds, terpenes, polysaccharides, and lipids. The therapeutic benefits of ginger are mainly attributed to phenolic compounds such as gingerol and shogaol [17]. The total phenolic content in the ginger extract was calculated based on the standard curve of gallic acid and represented as the gallic acid equivalent (GAE) per gram extract weight. The total phenolic content in the ginger extract was 43.32 ± 1.60 mg gallic acid/g extract (Table 1). As we know, 6-gingerol is the most abundant phenol in ginger [18]. The HPLC chromatogram of standard 6-gingerol showed a retention time of about 15.3 min, and the HPLC chromatogram of the ginger extract showed the major peak at a retention time of about 15.4 min (Figure 1). The result indicated that 6-gingerol is the main bioactive component in ginger extract. Quantification of the 6-gingerol in the ginger extract was achieved by using the standard curve. In this study, we found that the ginger extract contained 6-gingerol 219.7 ± 9.69 µg/mL (Table 1) or 22% 6-gingerol (*w/w*).

### 2.2. Antioxidant Activity of the Ginger Extract

The antioxidant activity of the ginger extract was determined based on the ability to scavenge ABTS^•+^ and DPPH^•^ radicals. In this study, the production of ABTS^•+^ was inhibited by the ginger extract and the positive control Trolox in a concentration-dependent manner (Figure 2a), as shown by IC50 of 1.43 ± 0.06 and 0.55 ± 0.07 mg/mL, respectively. Similarly, the production of DPPH^•^ radicals was inhibited by ginger extract and Trolox, with IC50 of 0.064 ± 0.002 and 0.012 ± 0.001 mg/mL, respectively (Figure 2b). The results indicated that ginger extract plays a role in the free radical scavenging activity.

### 2.3. Antihemolytic Activity of the Ginger Extract

The antioxidant activity of the ginger extract was determined by the inhibition of AAPH-induced RBC hemolysis. Ascorbic acid (AA) at a concentration of 0.1 mg/mL inhibited AAPH-induced RBC hemolysis (% inhibition = 95.31 ± 0.28%; Figure 3). In addition, the hemolysis of RBC was inhibited by the ginger extract at concentrations of 0.016–0.125 mg/mL. Interestingly, the ginger extract at concentrations of 0.125 and 0.063 mg/mL completely inhibited the hemolysis of RBC, as shown in Figure 3. This study suggested that the ginger extract exhibited potential antioxidant activity. 

### 2.4. The Effect of the Ginger Extract on Cell Viability

The toxicity of the ginger extract was studied via MTT assay in human hepatoma (Huh7) cells. The results in Figure 4 revealed that ginger extract at concentrations lower than 25 mg/mL was not toxic to the cells (% viability > 80%) over a 24 h treatment period. Likewise, the percent cell viability was more than 80% when the cells were treated with ginger extract at concentrations of less than 12.5 mg/mL for 48 h. Therefore, concentrations of ginger extract up to 12.5 mg/mL were applied in further experiments.

### 2.5. Intracellular Iron Chelating Activity of the Ginger Extract

Labile cellular iron (LCI) can induce ROS generation via the Fenton reaction. Thus, the iron-chelating activity of the ginger extract was determined in iron-loaded Huh7 cells. The level of LCI in the cell model increased significantly after treatment with ferrous ammonium sulfate (FAS) at concentrations of 0.2 mM (Figure 5). The level of LCI significantly decreased when the iron-loaded Huh7 cells were treated with the iron chelator, DFP at a concentration of 0.1 mM, and ginger extract at concentrations of 12.5–3.125 µg/mL (Figure 5). The results suggested that ginger extract exhibited antioxidant action by chelating the toxic iron in the cells.

### 2.6. The Intracellular Free-Radical Scavenging Activity of the Ginger Extract

The level of intracellular ROS was significantly increased when the Huh7 cells were loaded with FAS at a concentration of 0.2 mM (Figure 6). Treatment with tocopherol at a concentration of 200 µg/mL and ginger extract at a concentration of 12.5 µg/mL significantly diminished the intracellular ROS production (Figure 6). This study suggests that both tocopherol and ginger extract could scavenge the excessive intracellular ROS in iron-loaded Huh7 cells. Moreover, ginger extract at a concentration of 12.5 µg/mL was found to be more potent than the standard antioxidant compound, tocopherol.

### 2.7. Inhibitory Effect of the Ginger Extract on Lipid Peroxidation

The generation of intracellular ROS induces the lipid peroxidation of cell membrane components. Thiobarbituric acid reactive substances (TBARS), a product from the lipid peroxidation reaction, were identified in iron-loaded Huh7 cells. The results in Figure 7 revealed that treating Huh7 cells with FAS at a concentration of 0.2 mM markedly increased the level of TBARS compared with that of the no-treatment control. Treatment with tocopherol at a concentration of 200 µg/mL and ginger extract at concentrations of 3.125–12.5 µg/mL significantly reduced the levels of TBARS. The results suggest that ginger extract directly scavenges the ROS in iron-loaded Huh7 cells and subsequently prevents the cells from undergoing lipid peroxidation reactions.

## 3. Discussion

The body has no specific iron excretory mechanism, so the level of iron in the body must be tightly regulated. The presence of excess iron promotes reactive oxygen species (ROS) production, which is the primary cause of tissue damage and organ dysfunction [19]. Studies have revealed that chronic iron overload can induce the overproduction of free radicals, causing oxidative damage to many vital organs, particularly the liver, which is the main storage site of iron [20,21]. Numerous reports have demonstrated that ROS generation is the underlying mechanism of iron-induced oxidative tissue damage [21,22,23]. Patients who suffer from thalassemia and sickle cell anemia are vulnerable to iron overload, which is associated with hepatocellular injury [24]. In this study, the exogenous iron loading effectively increased the labile cellular iron (LCI) in the liver cells, which is correlated with the elevation of oxidative stress biomarkers, including intracellular ROS and lipid peroxidation products. We also found that chelation of LCI in the liver cells by ginger extract can prevent iron-induced oxidative liver damage by lowering the levels of intracellular ROS and lipid peroxidation products. This highlights the potential iron-chelating properties of ginger extract, which are partially associated with the major bioactive component in the ginger extract, 6-gingerol. It is well known that phenolic compounds can bind iron and provide antioxidant activity by inhibiting the Fenton-generated free radical [25]. Consistent with this study, a previous report demonstrated that the chelation of excess iron by Taxifolin, a naturally occurring flavonoid, can ameliorate iron-induced liver injury [26]. In addition, Inositol hexa phosphoric acid (IP6), a natural antioxidant of some leguminous plants, exhibited a hepatoprotective effect in an iron overload mice model by suppressing serum ferritin levels and, leading to a reduction in oxidative liver damage [27].

This study revealed that 6-gingerol is a major pharmacologically active phenolic compound found in ginger extract. The accumulated evidence confirmed that 6-gingerol demonstrated antiapoptotic, anticancer, antioxidant, and anti-inflammation properties [28,29,30,31]. In accordance with the previous findings, this study confirmed that the ginger extract exhibited free radical scavenging activity by inhibiting the production of ABTS^•+^ and DPPH^•^ radicals. Additionally, ginger extract can protect against free-radical attacks on the cell membrane of red blood cells. This finding is partially associated with the free radicals scavenging activity of 6-gingerol from ginger extract, which protects the cell membrane from the lipid peroxidation process. Furthermore, the activation of antioxidant enzymes has been documented as the underlying mechanism of 6-gingerol [32]. The previous report suggested that 6-gingerol potentially protected the liver from oxidative damage by enhancing the activity of various antioxidant enzymes such as superoxide dismutase (SOD) and glutathione-S-transferase (GST). In addition, 6-gingerol shows hepatoprotective effects by lowering the hepatic liver enzymes, including Aspartate transaminase (AST) and Alanine transaminase (ALT) [30]. The discovery of natural compounds that exhibited antioxidant action is an important strategy to overcome oxidative tissue damage and organ dysfunction. Thus, the free radical scavenging and iron-chelating activity of 6-gingerol from ginger extract can be attributed to the various pharmacological properties that protect the liver from oxidative damage.

## 4. Materials and Methods

### 4.1. Chemical Reagent

2,2′-Azinobis 3-ethylbenzothiazoline-6-sulphonate (ABTS), 2′,7′-dich lorodihydrofluorescein diacetate (DCFH-DA), dimethyl sulfoxide (DMSO), 2,2-diphenyl-1picrylhydrazyl-hydrate (DPPH), Dulbecco’s modified Eagle’s medium (DMEM), fetal bovine serum (FBS), Ferroorgane, Ferrous ammonium sulfate (FAS), α-tocopherol (TC), and thiobarbituric acid (TBA) were purchased from Thermo Fisher Scientific, Inc., Waltham, MA, USA. Additionally, 3-(4,5Dimethylthiazol-2-yl)-2,5-diphenyltetrazolium bromide (MTT) and phosphate-buffered saline (PBS) at a pH of 7.0 were purchased from Sigma–Aldrich Inc., St. Louis, MO, USA.

### 4.2. Plant Extraction

Fresh ginger (*Zingiber officinale* Roscoe) was purchased from the Thai Royal Project shop, Chiang Mai, Thailand, then washed and dried at 50 °C for 24 h. The dried ginger was powdered and extracted using absolute ethanol. The ginger solution was shaken using a refrigerated incubator shaker (Amerex Gyromax 737R, Marshall Scientific Co., Ltd., Hampton, VA, USA) at 200 rpm at room temperature for 24 h. The supernatant was collected and filtered with filter paper. Finally, the ethanol extract was evaporated using a rotary evaporator (Hei-VAP Expert Control, Heidolph Instruments GmbH & Co., Schwabach, Germany). The ginger extract was reconstituted in absolute ethanol. During the cell culture experiments, DMSO was used to prepare the stock solution and culture medium was used to dilute the stock into various concentrations.

### 4.3. Chemical Composition Determination

#### 4.3.1. Total Phenolic Content

The Folin–Ciocalteu reagent was used to determine the total phenolic content of the ginger extract. Briefly, the ginger extract (0.125–1 mg/mL) 20 µL was mixed with 80 µL Folin–Ciocalteu reagent and 100 µL 7.5% sodium carbonate. The reaction mixture was incubated in the dark at room temperature for 2 h. Then, the absorbance of the mixture was measured at 765 nm. Gallic acid was used as a standard. The total phenolic content was expressed as gallic acid equivalents (GAEs).

#### 4.3.2. Determination of 6-Gingerol 

6-gingerol is a chief pharmacologically active phenolic compound in ginger and plays a crucial role in various biological activities [30]. The 6-gingerol was identified using high-performance liquid chromatography (HPLC). The C-18 reverse phase column was used to identify 6-gingerol in the ginger extract. The mobile phase consisted of solvent A containing 0.05% (*v*/*v*) ortho–phosphoric acid and solvent B containing methanol. The 6-gingerol was isocratically eluted with solvent A: solvent B (60:40 *v*/*v*) at a flow rate of 1 mL/min. The sample was detected using a UV detector at the wavelength of 280 nm. The 6-gingerol content in the ginger extract was calculated based on the standard curve of 6-gingerol. 

### 4.4. Antioxidant Activity Measurement

#### 4.4.1. DPPH Assay

The radical scavenging activity of the ginger extract was determined via DPPH assay. The concentrations of ginger extract at 0.03125–1 mg/mL and Trolox at 0.002–0.5 mg/mL (0.2 mL) were incubated with 0.4 mM DPPH^•^ solution (0.2 mL) in the dark for 30 min at room temperature. The absorption of the reaction mixture was monitored photometrically at 517 nm. The antioxidant activity was expressed as percent inhibition of DPPH^•^ production, which was calculated using the following equation: 

Inhibition of DPPH^•^ production (%) = [(A − B)/A] × 100, where A is the absorbance of the control and B is the absorbance of the sample. IC 50 is the concentration of ginger extract that can inhibit 50 percent of DPPH radicals. This was calculated from a scatter graph between the concentration of ginger extract and the percentage of inhibition.

#### 4.4.2. ABTS Assay

The ABTS radical cation (ABTS^•+^) was generated by combining the solution of 7 mM ABTS and 2.45 mM potassium persulfate in the dark at room temperature for 12–16 h. The resulting ABTS^•+^ was diluted with DI water to achieve an absorbance of 0.70 (±0.02) at 734 nm. For the radical scavenging reaction, the concentration ranges of 0.03125–8 mg/mL of ginger extract and 0.078–2.5 mg/mL of Trolox (10 µL) were mixed with ABTS^•+^ solution (1 mL) and incubated in the dark at room temperature for exactly 6 min. The absorbance was monitored at 734 nm. The antioxidant activity was expressed as percent inhibition of ABTS^•+^ production, which was calculated using the following equation: 

Inhibition of ABTS^•+^ production (%) = [(A − B)/A] × 100, where A is the absorbance of the control and B is the absorbance of the sample. The IC50 of ginger extract for ABTS^•+^ radicals inhibition was calculated based on the graph plotted between the concentration of ginger extract and the percentage of inhibition.

#### 4.4.3. AAPH Assay

AAPH is a source of free radicals. AAPH-induced hemolysis was performed according to Paradee et al. (2019) [33]. Briefly, the red blood cells were washed 3 times and suspended in PBS solution pH 7.4. Then, the 20% suspension of red blood cells (0.1 mL) was mixed with 200 mM AAPH solution (0.1 mL) followed by the ginger extract at 0.015–1 mg/mL and ascorbic acid at 0.1 mg/mL (0.1 mL). The reaction mixture was incubated at 37 °C for 3 h. After incubation, 0.4 mL of PBS solution was added to the reaction mixture, followed by centrifugation at 3000× *g* for 10 min. The absorbance was measured at OD 540 nm. The antihemolytic activity was expressed as percent inhibition of AAPH-induced hemolysis, which was calculated using the following equation:

Inhibition of AAPH-induced hemolysis (%) = [(A − B)/A] × 100, where A is the absorbance of the control and B is the absorbance of the sample.

### 4.5. Cell Culture

Human hepatoma (Huh7) cells were cultured in Dulbecco’s minimal essential medium (DMEM) supplemented with 10% FBS, penicillin (100 IU/mL), and streptomycin (100 µg/mL) at 37 °C in a humidified atmospheric incubator containing 5% CO_2_. Cells were harvested at 80–90% confluence.

### 4.6. Cytotoxicity Assay

The toxicity of ginger extract in Huh7 cells was established using the MTT assay. Huh7 cells were seeded in 96-well culture plates and treated with various concentrations of ginger extract (final concentration of 0–50 µg/mL) for 24 and 48 h. The cells were then incubated with MTT (5 mg/mL) for 4 h. Finally, the blue-colored formazan product was extracted with 0.1 mL of dimethyl sulfoxide (DMSO), and the absorbance at 540 nm and 630 nm was monitored. The cell viability was expressed in comparison with untreated cells (100% viability).

### 4.7. Determination of Oxidative Stress Markers

#### 4.7.1. Intracellular ROS

The intracellular ROS was measured using the previously described dichlorodihydrofluorescein diacetate (DCFH-DA) assay [34]. Huh7 cells were seeded in 96-well culture plates (1 × 10^4^ cells/well). The cells were treated with various concentrations of ginger extract (0–12.5 µg/mL) and tocopherol (200 µg/mL) at 37 °C for 24 h. Afterward, 20 µM of DCFH-DA was added and incubated in the dark at 37 °C for 30 min. The excess DCFH-DA was removed by washing with PBS solution. Then, the cells were challenged with 0.2 mM ferrous ammonium sulfate (FAS) for 1 h to generate the free radicals. The fluorescent intensity of the dichlorodihydrofluorescein (DCF) product was measured based on the excitation wavelength at 485 nm and the emission wavelength at 535 nm. The result was expressed as percent fluorescent intensity (% FI).

#### 4.7.2. Lipid Peroxidation

Thiobarbituric acid reactive substances (TBARS) are a product of the lipid peroxidation process. In the assay, TBARS was measured using the method described by Yang et al. [35]. Briefly, the Huh7 cells were seeded in 6-well culture plates at a density of 8 × 10^5^ cells/well. The cells were pretreated with various concentrations of ginger extract (0–12.5 µg/mL) and tocopherol (200 µg/mL) at 37 °C for 24 h, followed by exposure to 0.2 mM FAS for 2 h. Afterward, the cells were detached using trypsin/EDTA and washed with PBS solution twice. Then, the cells were lysed via sonication. The cell lysate (0.1 mL) was incubated with 1% phosphoric acid (0.3 mL) and 0.67% thiobarbituric acid (TBA; 0.1 mL). The reaction mixture was boiled at 95 °C for 1 h. After cooling, the TBARS product was extracted with n-butanol (0.4 mL). The absorbance of the organic phase was monitored photometrically at OD 535 nm. The cell lysate protein was determined using the Bradford assay. The result was analyzed based on the standard curve of 1,1,3,3-tetra methoxy propane (TMP). 

### 4.8. Determination of Labile Cellular Iron

Labile cellular iron (LCI) is an intracellular transient redox-active iron that can induce the production of ROS via the Fenton reaction. The level of LCI was measured via FerroOrange assay (Dojindo, Japan). Briefly, The Huh7 cells were seeded in 96-well culture plates at a density of 1 × 10^4^ cells/well and pretreated with various concentrations of ginger extract (0–12.5 µg/mL) and DFP (100 µM) at 37 °C for 24 h. Then, the cells were treated with 0.2 mM FAS for 1 h to load iron into the cells. Afterward, iron-loaded cells were washed twice with Hank’s Balanced Salt Solution (HBSS) to remove the extracellular iron. Finally, 1 mM FerroOrange was added and incubated in the dark at 37 °C for 30 min. The fluorescence intensity was monitored at the excitation wavelength of 556 nm and the emission wavelength of 615 nm.

### 4.9. Ethical Approval

The protocol for the AAPH assay, which uses human red blood cells, was approved by the Research Ethics Committee of the Faculty of Medicine, Chiang Mai University (Study Code: BIO-2564-08002, Research ID: 8002).

### 4.10. Statistic

Data were analyzed using GraphPad Prism version 8.0 (GraphPad Prism Software, San Diego, CA, USA) and are expressed as the mean ± SD, n = 3. Statistical significance was analyzed using a one-way analysis of variance with post hoc Tukey–Kramer, in which *p* < 0.05 was considered a significant difference.

## 5. Conclusions

The present study highlights the protective effects of ginger extract against iron-induced liver injury. Our findings suggest that high iron levels or iron overload can cause oxidative damage to many vital organs, particularly the liver, by inducing free-radical production via the Fenton reaction. Thus, the chelation of excess iron by ginger extract would overcome the consequences of iron-induced oxidative liver damage in iron overload patients.

## Figures and Tables

**Figure 1 plants-12-02936-f001:**
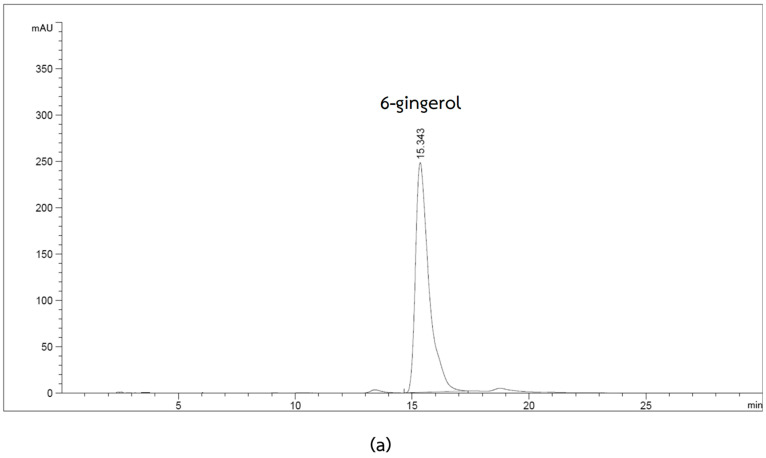
HPLC chromatogram of standard 6-gingerol (**a**) and ginger extract (**b**). The peaks were detected by a UV detector at 280 nm. The content of 6-gingerol in the ginger extract was identified via comparison with the retention times of the standard 6-gingerol.

**Figure 2 plants-12-02936-f002:**
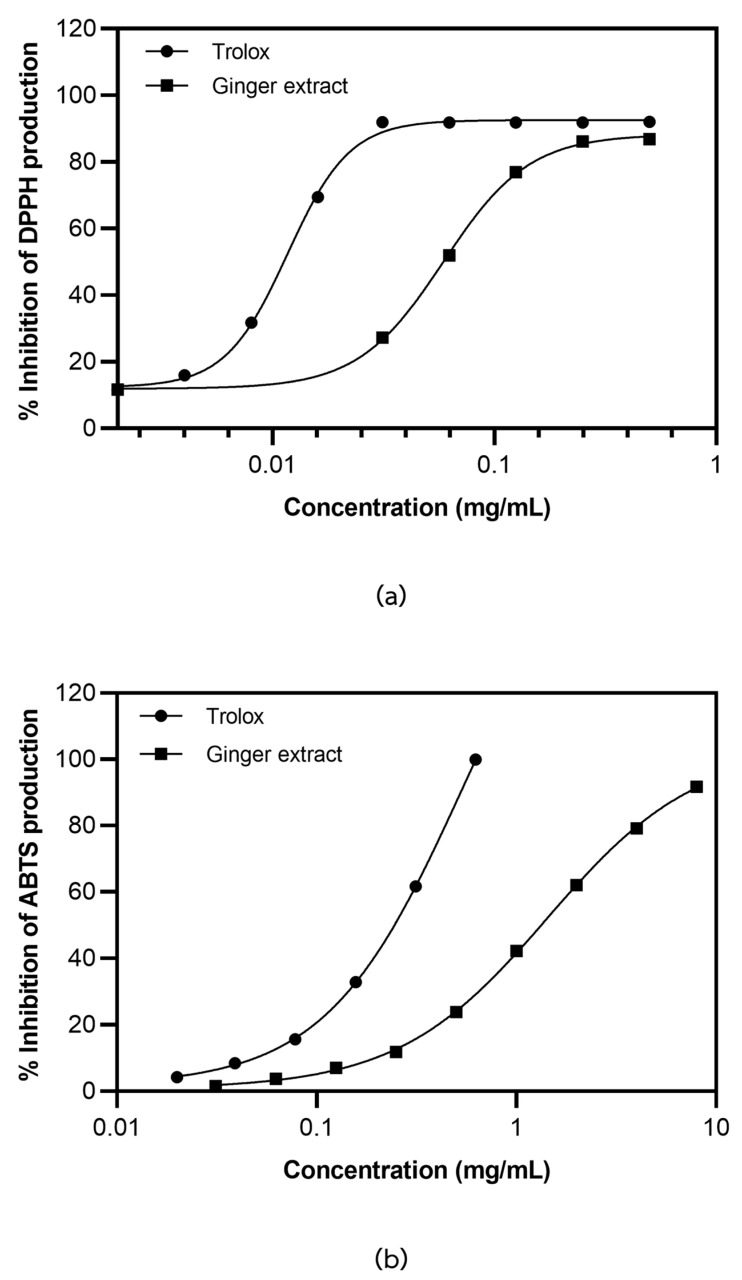
The scavenging activity of ginger extract to ABTS^•+^ (**a**) and DPPH^•^ (**b**) radicals production. The data were obtained from three independent experiments and expressed as mean ± SD.

**Figure 3 plants-12-02936-f003:**
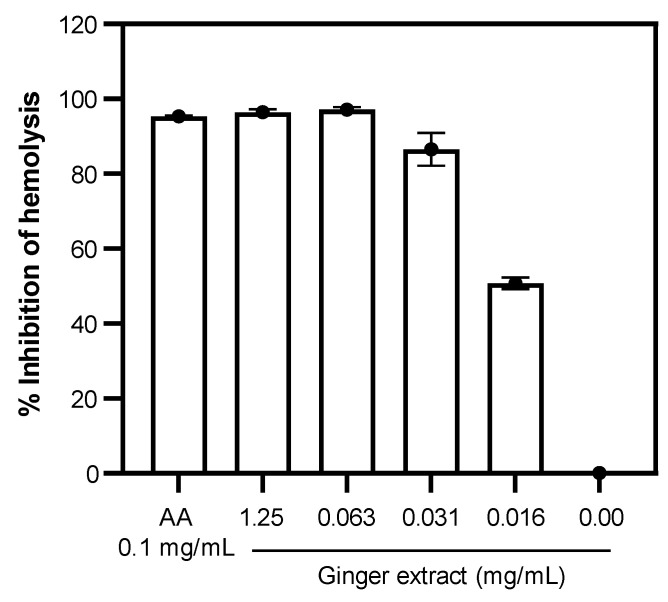
The inhibition of AAPH-induced hemolysis by ginger extract and ascorbic acid (AA). Data were obtained from three independent experiments and expressed as mean ± SD.

**Figure 4 plants-12-02936-f004:**
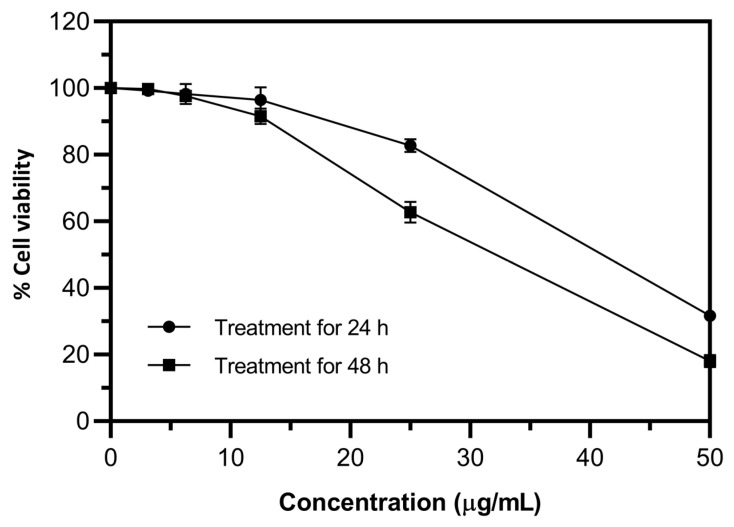
Viability of human hepatoma (Huh7) cells treated with various concentrations of ginger extract for 24 h (●) and 48 h (■). Data were obtained from three independent experiments and expressed as mean ± SD.

**Figure 5 plants-12-02936-f005:**
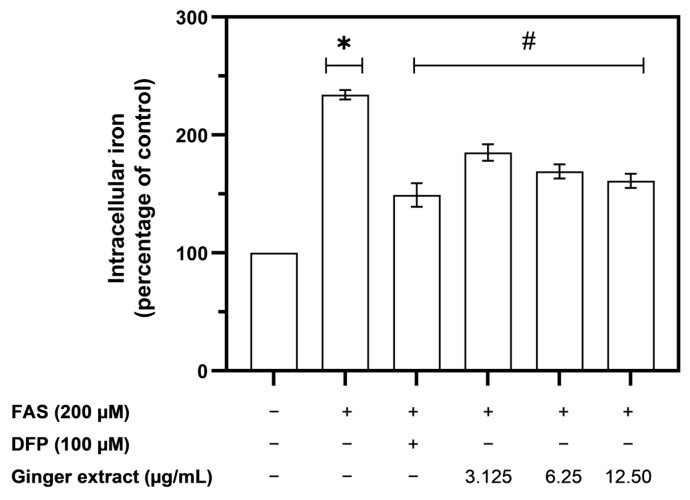
The effect of ginger extract and DFP on the level of labile cellular iron (LCI) in ironloaded human hepatoma (Huh7) cells. Data were obtained from three independent experiments and are expressed as the mean ± SD. * *p* < 0.05 when compared with no-treatment control; # *p* < 0. 05 when compared with FAS treatment.

**Figure 6 plants-12-02936-f006:**
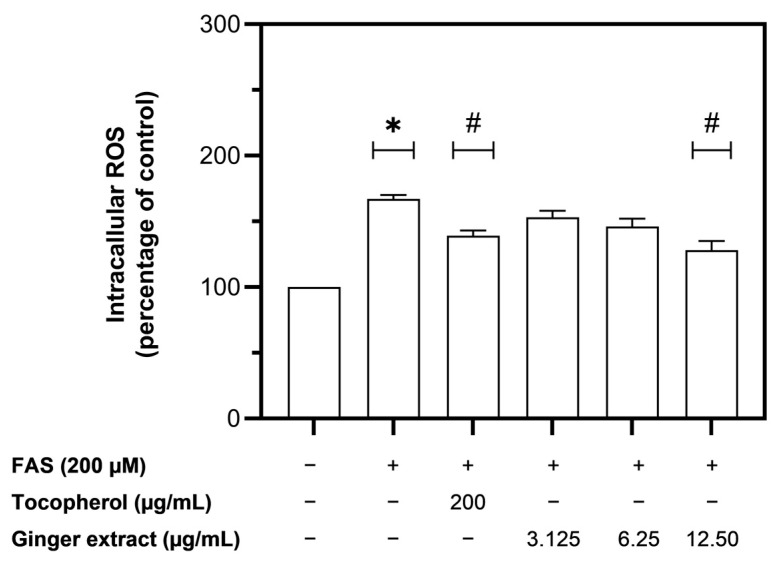
The level of intracellular ROS in iron-loaded Huh7 cells when treated with tocopherol and ginger extract. Data were obtained from three independent experiments and expressed as mean ± SD. * *p* < 0.05 compared with no-treatment control; # *p* < 0. 05 compared with FAS treatment.

**Figure 7 plants-12-02936-f007:**
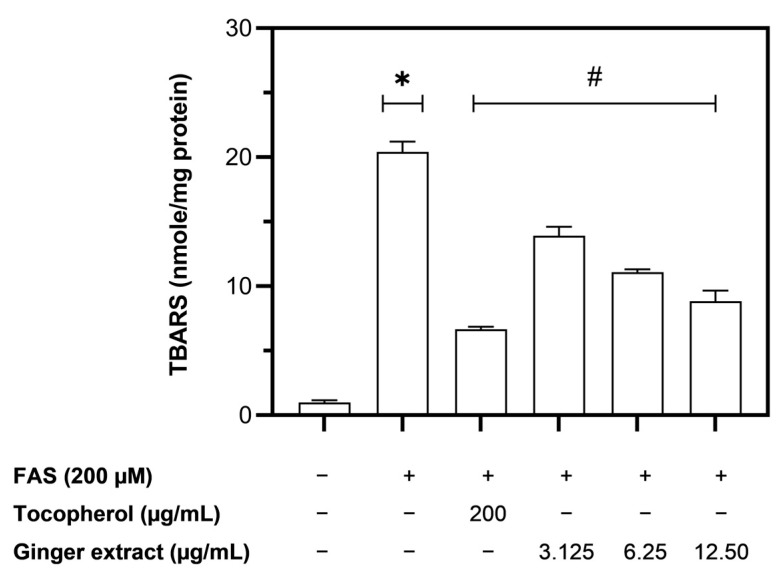
The level of TBARS in iron-loaded Huh7 cells treated with tocopherol and ginger extract. Data were obtained from three independent experiments and expressed as mean ± SD. * *p* < 0.05 compared with no-treatment control; # *p* < 0.05 compared with FAS treatment.

**Table 1 plants-12-02936-t001:** Total phenolic content (TPC), 6-gingerol, and antioxidant activities of ginger extract and positive control. The data were obtained from three independent experiments and are expressed as mean ± SD.

Extract/Control	TPC (mg GAE/g Extract)	6-Gingerol (µg/mL)	ABTS IC50 (mg/mL)	DPPH IC50 (mg/mL)
Ginger extract	43.32 ± 1.60	219.7 ± 9.69	1.43 ± 0.06	0.064 ± 0.002
Trolox	-	-	0.55 ± 0.07	0.012 ± 0.001

## Data Availability

Not applicable.

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
