# Peer review of "Iron Chelation Property, Antioxidant Activity, and Hepatoprotective Effect of 6-Gingerol-Rich Ginger (Zingiber officinale) Extract in Iron-Loaded Huh7 Cells"

_plants, 2023, doi:10.3390/plants12162936_

Round 1

Reviewer 1 Report

1. The subheading 4.4 should be revised. This section is about ginger extracts preparation. The current subheading is confusing

2. Where did the authors obtain or purchase the ginger?

3. Did the authors grind their ginger before extraction with ethanol?

4. Under what conditions was the ethanol evaporated?

5. What were the indicated concentrations? It is better to state the concentrations

6. The spelling of Folin C should be revised

7. At what range of gallic acid concentration was the standard curve linear? What were the final units of TPC?

8. More information should be added on the HPLC method for determining 6-gingerol

9. What were the various concentrations of ginger extract? (line 273)

10. How was the absorbance monitored?  (line 274)

11. Why did the authors used DI to adjust the absorbance of ABTS not alcohol?

12. The authors should avoid generalizing important information in the methodology, for instance the concentrations of ginger extracts used.

13. Why did the authors not use standard curves to determine the antioxidant activity

14. The use of FAS, DFP, tocopherol, trolox to compare the efficiency of ginger extracts is not included in the methodology

15. The determination of ABTS and DPPH IC50 is not included in the methodology

The quality of English should be improved.

Author Response

Please find an attached file.

Reviewer 2 Report

The manuscript describes how ginger extract can diminish the oxidative process induced by iron excess. The different experiments are well done, the discussion is well documented and also the conclusions are agree to the objetives of the research, but there are some details that the authors should consider to increase the quality of the paper. One of them is the units to express concentration of total phenolic compounds. It should be expressed in g/g dry weight. The another point is related to 6-gingerol, because the authors pointed out that this compound is responsible of the biological activity found. If the authors would have studied the pure compound, their assumptions could be true.

Author Response

Please find an attached file.

Round 2

Reviewer 1 Report

The authors have adequately addressed the comments. The manuscript may be considered for publication.